# A Non-Isocyanate Route to Poly(Ether Urethane): Synthesis and Effect of Chemical Structures of Hard Segment

**DOI:** 10.3390/polym14102039

**Published:** 2022-05-16

**Authors:** Ziyun Shen, Liuchun Zheng, Danqing Song, Yi Liu, Chuncheng Li, Jiajian Liu, Yaonan Xiao, Shaohua Wu, Tianbo Zhou, Bo Zhang, Xuedong Lv, Qiyong Mei

**Affiliations:** 1Beijing National Laboratory for Molecular Sciences, Key Laboratory of Engineering Plastics, Institute of Chemistry, Chinese Academy of Sciences (ICCAS), Beijing 100190, China; shenziyun15@iccas.ac.cn (Z.S.); liuchunzheng@tiangong.edu.cn (L.Z.); ljj6609891@iccas.ac.cn (J.L.); ynxiao@iccas.ac.cn (Y.X.); wushaohua12@iccas.ac.cn (S.W.); zhangbo@iccas.ac.cn (B.Z.); 2School of Textile Science and Engineering, Tiangong University, Tianjin 300387, China; 2120010033@tiangong.edu.cn (D.S.); yiliuchem@whu.edu.cn (Y.L.); zhoutianbo0816@163.com (T.Z.); lxd998520@163.com (X.L.); 3Department of Neurosurgery, Changzheng Hospital, Naval Medical University, Shanghai 200003, China

**Keywords:** non-isocyanate, poly(ether urethane), hard segment

## Abstract

A series of non-isocyanate poly(ether urethane) (PEU) were prepared by an environmentally friendly route based on dimethyl carbonate, diols and a polyether. The effect of the chemical structure of polyurethane hard segments on the properties of this kind of PEU was systematically investigated in this work. Polyurethane hard segments with different structures were first prepared from hexamethylene di-carbamate (BHC) and different diols (butanediol, hexanediol, octanediol and decanediol). Subsequently, a series of non-isocyanate PEU were obtained by polycondensation of the polyurethane hard segments with the polyether soft segments (PTMG2000). The PEU were characterized by GPC, FT-IR, ^1^H NMR, DSC, WAXD, SAXS, AFM and tensile testing. The results show that the urea groups generated by the side reaction affect the degree of crystallization of hard segments by influencing the hydrogen bonding of the hard segments molecular chains. The degree of hard segment crystallization, in turn, affects the thermal and mechanical properties of the polymer. The urea group content is related to the carbon chain length of the diol used for the synthesis of hard segments. When butanediol is applied to synthesize hard segment, the hard segment of the resulting PEU is unable to crystallize. Therefore, the tensile strength and modulus of elasticity of butanediol-based PEU is lowest among three, though it possesses the highest urea group content. When longer octanediol or decanediol is applied to synthesize the hard segment, the hard segments in the resulting polyether-based polyurethane are crystallizable and the resulting PEU possesses higher tensile strength.

## 1. Introduction

Polyurethane (PU) is a block copolymer composed of soft and hard segments. Polyurethane materials are widely used in the manufacture of foams, elastomers, fibers, coatings, synthetic leather, adhesives, surfacing materials and medical materials due to their excellent elasticity, toughness abrasion and weather resistance [1,2]. Therefore, it has a very good prospect in the fields of transportation, construction, light industry, textile, electromechanics, aviation, medical and health care [3,4,5]. Generally, the type and proportion of hard and soft segments in polyurethane affects thermal and mechanical properties of polyurethane materials substantially. According to the type of synthetic soft segment materials, polyurethane can be divided into polyether polyurethane (PEU), polyester polyurethane and polycarbonate polyurethane. PEU is a polyurethane material whose soft segment is polyether diol. The polyether polyol repeating unit, with low cohesion energy and good ability to rotate, endows the PEU material good low temperature flexibility and excellent hydrolysis resistance [6].

Usually, such polymers are mainly synthesized from isocyanates and polyols, and the technology is very mature and the products possess excellent properties [7]. However, with the improvement of environmental protection requirement and the development of green technology, many problems have been exposed to the traditional production and processing of polyurethane: diisocyanate raw materials are volatile, highly toxic and harmful to humans and environment [8,9]. Moreover, highly poisonous phosgene is used as the main material for industrial production of diisocyanate [10]. Additionally, diisocyanate are highly susceptible to moisture in the air, which affects the processes of polyurethane preparation, transport, and storage of isocyanates [11]. Therefore, the investigation of a green non-isocyanate process for polyurethane to replace the conventional synthetic route is crucial for academic research and industrial development.

Several researchers have reported their exploratory work on this topic. Generally, there were two non-isocyanates routes for the synthesis of polyurethanes [12,13,14,15,16,17]. The first one is the preparation of polyurethane by addition reaction of binary cyclic carbonate and binary amine. The reaction temperature of this route was generally below 100 °C, the products did not contain volatile organic compounds, and the reaction system is not sensitive with moisture. However, most of these reactions required the use of large amounts of polar organic solvents, such as DMSO, DMF, etc., and the synthesis process is long and time-consuming. Leitsch et al. [18]. prepared a series of polyether-based PHUs using cyclic carbonate-capped polyethers and dibasic amines and small molecules of dibasic cyclic carbonate as raw materials. The presence of hydroxyl groups in the molecular structure of the obtained PHU significantly affects the micro-phase separation of polyether-based polyurethanes. Thermoplastic polyurethanes prepared using conventional methods with polyethylene glycol as the soft segment have a two-phase structure. The corresponding PHU, however, is a single-phase copolymer, due to the presence of hydroxyl groups, that flows under gravity and cannot be used as an elastomer. Zhao Jingbo’s group [19] prepared diurethane diols by reacting vinyl carbonate with hexanediamine, and then condensed with polyethylene glycol to obtain polyether-type polyurethanes with a number average molecular weight up to 31,000 g/mol, tensile strength in the range of 5–24 MPa and elongation at break in the range of 0.9–1388%. However, none of the polymers showed a combination of good strength and flexibility.

The second method was the preparation of polyurethanes by ester exchange reaction of di-carbamates with diols. This method does not use solvents, and was a green synthetic route derived from green materials [20,21,22]. The dicarbamates are mainly synthesized from cyclic carbonate and dimethyl carbonate (DMC). DMC is a widely used non-toxic reagent, which does not require special protective measures for irritation or mutagenic effects caused by contact or inhalation [23,24,25]. Kebir et al. [26] prepared polyether-based polyurethanes from dimethyl carbonate, using polyethylene glycol as the soft segment and butylene glycol as the chain extender. The molecular weight (7500–14,800 g/mol) and melting point (38–48 °C) of the obtained polyurethanes are not high enough, making it difficult to meet the use application requirements. Researchers have focused on the mechanism of the melt polycondensation reaction of the di-carbamate monomer with the diol, the control of the polycondensation reaction conditions, and the selection of catalysts [17,27,28,29,30,31]. Our previous works have investigated the difference of the chemical structure, thermal properties and mechanical properties between the dimethyl carbonate based PEU and traditional PEU [32]. However, no systematic reports on the structure–property relationships of polyether-based polyurethanes prepared by the DMC route have been published. Furthermore, there are few discussions on their application prospects or reports on whether their mechanical properties meet the practical needs.

In this work, DMC based non-isocyanate polyether-based polyurethanes with various hard segment have been prepared. Influence of chemical structure of hard segment on the properties of non-isocyanate polyether-based polyurethanes has been investigated for the first time. The properties of the obtained polymers were systematically characterized by ^1^H NMR, FT-IR, GPC, DSC, WAXD, SAXS, AFM and tensile testing.

## 2. Materials and Methods

### 2.1. Materials

DMC was used as received from Shandong Shida Shenghua Chemical (Shandong, China). 1,6-hexanediamine (HDA), butanediol (BDO), hexanediol, octanediol, decanediol were bought from J&K Chemicals (Beijing, China) and used without purification, tetrabutyl titanate (TBT) and K_2_CO_3_ were purchased from Beijing Chemical Reagents Corp. (Beijing, China) and used without purification. Sodium acetate, and deuterated methyl sulfoxide were bought from Beijing InnoChem Science & Technology Corp (Beijing, China) and used without purification. Poly(tetramethylene ether) glycol (PTMG, *M*_n_ = 2000 g/mol) was purchased from Aladdin (Shanghai, China) and used without purification.

### 2.2. Synthesis of Bis-1,6-hexamethylencarbamate (BHC)

Bis-1,6-hexamethylencarbamate (BHC) was synthesized from DMC and HDA according to the reference, using DMC as the solvent [33]. Crude yield: 84%. ^1^H NMR data (400 MHz, DMSO-*d*_6_, δ in ppm): 7.06 (br, 2H, NH); 3.50 (s, 6H, OCH_3_); 2.97–2.92 (q, 4H, CH_2_NH); 1.38–1.35 (m, 4H, CH_2_CH_2_NH); 1.24–1.20 (m, 4H, CH_2_CH_2_CH_2_NH).

### 2.3. Synthesis of Polyurethane Diol (PUDL)

There are two steps for synthesis of PUDL. The first step is the transesterification reaction. Nitrogen was passed into the four-necked flask with a condenser and mechanical stirring device. When the temperature reached 120 °C, BHC and diol (BDO or hexanediol or octanediol) (molar ratio BHC to diol = 1/2) were added and mixed uniformly, then the catalyst potassium carbonate (0.1 wt% of BHC mass) was fed. The reaction system was slowly heated to 165 °C and maintained for 5 h until no more methanol distils out. The second step was polycondensation. Vacuum was applied to the reaction system until the pressure dropped below 20 Pa and kept for 1 h to distil out the excess glycol. As the boiling point of decanediol is around 300 °C at 101 kPa, 192 °C at 2.67 kPa, 170 °C at 1.07 kPa, it could be distilled at 170 °C at 20 Pa during polycondensation. Other diols with lower molecular weight have lower boiling points than decanediol, and they could be distilled at 170 °C at 20 Pa during polycondensation. Therefore, the hydroxyl-terminated prepolymer PUDL was obtained. The synthesized PUDL was named as PUDL4, PUDL6, PUDL8, and PUDL10 according to the number of carbon atoms of the diol. The chemical structure and number average molecular weight of the prepolymer was characterized by hydrogen nuclear magnetic spectroscopy.

^1^H NMR data (400 MHz, DMSO-*d*_6_, δ in ppm): 6.45 (–NHC(O)O–); 4.10(–OC(O)OCH_2_–); 3.97–3.94 (–NHC(O)OCH_2_–); 3.38 (–CH_2_O–) 3.02–2.98 (–CH_2_NH–); 1.57–1.21 (–CH_2_–).

### 2.4. Synthesis of Polyether Polyurethane (PEU)

After the air in the reaction kettle was replaced with nitrogen, the prepolymer and polyether diol PTMG2000 (dried at 110 °C under vacuum for one hour before use) were fed into the kettle, and the temperature was increased to 170 °C. After the reactants were melted and mixed well, tetrabutyl titanate (0.3 wt%), the condensation catalyst, was added and stirred well. The pressure of the reaction system was then slowly decreased to 20 Pa. The polycondensation reaction was carried out at 170 °C for 4–8 h under vacuum to obtain non-isocyanate polyether-based polyurethane (PEU). The samples are named as PEU4, PEU6, PEU8, and PEU10, and the numbers represent the number of carbon atoms of the straight-chain diols used for the preparation of the polyurethane hard segment. ^1^H NMR data (400 MHz, DMSO-*d*_6_, δ in ppm): 6.45 (–NHC(O)O–); 4.10(–OC(O)OCH_2_–); 3.97–3.94 (–NHC(O)OCH_2_–); 3.38 (–CH_2_O–) 3.02–2.98 (–CH_2_NH–); 1.57–1.21 (–CH_2_–).

### 2.5. Characterizations

The chemical structures of PEU were characterized by NMR spectrometer (Bruker DMX-400, Bruker Instruments, Switzerland) at a concentration of 10 mg/mL at 100 °C, using DMSO-*d*_6_ as the solvent. A Thermo Nicolet Avatar 6700 FT-IR equipped with an attenuated total reflectance device (Smart Orbit, Thermo Scientific, Rochester, MN, USA) was employed. The samples were scanned from 400 to 4000 cm^−1^ with a resolution of 4 cm^−1^ at room temperature.

The number-average molecular weight (*M*_n_), weight-average molecular weight (*M*_w_) and dispersity (Đ) were determined by GPC (Waters 2414, Waters, Milford, MA, USA). The measurements were taken at 45 °C, using DMF as eluent at a flow rate of 1.0 mL/min. The *M*_n_ was calculated by using a calibration curve with monodisperse polystyrene as standards.

A TA instrument (Perkin-Elmer Pyris Diamond DSC Q2000, TA Instruments, New Castle, DE, USA) was used for the thermal properties characterization. The samples (3–5 mg) were first heated to 180 °C under N_2_ atmosphere and held there for 5 min to eliminate the thermal history. Then, the samples were cooled to −70 °C at a rate of 200 °C/min. Thereafter, the samples were reheated to 180 °C at a rate of 10 °C/min and held there for 5 min, then cooled to −120 °C at the same rate to obtain heating and cooling curves.

Wide-angle X-ray analysis (WAXD) and small-angle scattering (SAXS) of PEU were measured by an X-ray diffractometer (Xeuss 2.0 system of Xenocs, Xenocs, Grenoble, France), and diffraction patterns were collected by a detector (Pilatus 300 K, DECTRIS, Switzerland) with Cu/Kα as the source (λ = 0.154 nm). The distance between the sample and the detector was 140.2 mm for the wide angle tests and 2500 mm for the small angle test. The time for each sample collection was 30 min. All samples were heated to 180 °C at a heating rate of 10 °C/min before testing under nitrogen atmosphere, held for 5 min and then naturally cooled to room temperature. The natural cooling rate was about 3 °C/min.

Atomic force microscope (AFM) testing was performed on a Digital Instrument Multimode Nanoscope in intelligent mode. The sample was dissolved in formic acid with a concentration of 1 wt%, and the solution was dropped on a silicon wafer to make a thin film sample of 30–50 µm thickness. The samples were then heated to 180 °C under nitrogen atmosphere at a heating rate of 10 °C/min, held there for 5 min and then cooled naturally to room temperature before AFM testing.

Mechanical properties of PEU were tested on a universal tester (Instron 1122, Instron, Norwood, MA, USA), and the tensile test was performed according to ISO 527–5A. The samples were molded on a small injection molding machine (Haake Minijet, Thermo Fisher, Karlsruhe, Germany) at 190 °C, the mold temperature is 45 °C and the sample size was “75 × 4 × 2” mm. The melt cavity temperature was 180 °C and the mold temperature was 30 °C. During stretching, the moving speed of the fixture beam was 50 mm/min, and the stretching data was taken as the average of five measurements.

## 3. Results and Discussion

### 3.1. Synthesis and Structural Characterization of PEUs

In order to further investigate the structure–property relationship of polyurethane, the reaction of hexamethylene di-carbamate monomer (BHC) with straight-chain diols of different carbon atomic numbers was used to obtain a series of PUDLs (butanediol, hexanediol, octanediol and decanediol), and the molecular weight of PUDL4 was calculated to be 840 g/mol; the molecular weight of PUDL6 was 804 g/mol; the molecular weight of PUDL8 was 669 g/mol; and the molecular weight of PUDL10 was 670 g/mol. The soft segment of polyether-based polyurethanes cannot self-condense under high temperature and high vacuum conditions and can only react with the hard segment PUDL in a condensation reaction, so the average length of the soft segment PTMG in the polymer does not change and is 2000 g/mol. When the soft segment is condensed with the hard segment, molecular weight growth can only be achieved by removing the small diol from the PUDL end of the hard segment. In the polyether polyurethanes prepared, both the small diol and the PTMG can only be linked to the BHC molecule, so [BHC] = [PTMG] + [Diol] in the polymer. Only small molecules of the diol are removed and distilled out during the polycondensation, the amount of BHC and PTMG in the system does not change. Therefore, it is possible to calculate the feeding mass ratio accordingly to ensure that the polyether based polyurethane prepared has a soft segment mass fraction of 70%. The following is using condensation of PUDL4 and PTMG2000 as an example to illustrate the calculation of the feeding mass.

The calculation of molar ratio of BHC, PTMG, and BDO in the molecular chain of the polyether polyurethane was based on the formula $\left(m_{\mathrm{PTMG}}–m_{\mathrm{OH}}\right)/m_{\mathrm{total}}=70\%$, [BHC] = [PTMG] + [Diol], which gives [BHC]/[BDO]/[PTMG] = 3.48/2.48/1. If 20 g of PTMG2000 was fed into the reactor, the hard segment should contain $20\times \frac{3.48}{2000}=0.0348\mathrm{mol}$ of BHC. The polymerization degree of polyurethane hard segment of PUDL4 was calculated to be 2.91 from the NMR results, and the molecular weight was 840 g/mol, so the fed hard segment should be 0.0348 × 840/2.91 = 10.04 g.

The preparation route for polyether-based polyurethanes was shown in Figure 1. The conditions and reaction reagents were identical for the four samples, except for diols used in the preparation of the hard segments, and the mass fraction of the soft and hard segments was also same. The purpose of this study was to examine the effect of the carbon chain length of polyurethane hard segment diols on the crystallization of hard segments and the properties of polyether-based polyurethanes. In order to achieve the polymers with comparable molecular weight, the polycondensation reaction was stopped when the stirring rate is 10 r/min under a voltage of 125 V. The polyurethanes were named as PEU4, PEU6, PEU8 and PEU10 according to the number of carbon atoms of the chain extender diol. The chain segment composition as well as molecular weights of PEU are listed in Table 1. The molecular weights of all four samples were above 50,000 g/mol and the molecular weight distribution was in the range of 1.7–2.3.

The structures of polyether polyurethanes were characterized using ^1^H NMR and IR spectra. Figure 2 showed the ^1^H NMR spectra of the four hard segment prepolymers and their attributions, and Figure 3 showed the ^1^H NMR spectra of the four corresponding polyether-type polyurethanes and their attributions. Compared with polyester-based polyurethanes and polycarbonate polyurethanes, polyether-based polyurethanes had poor solubility and were difficult to dissolve in deuterated DMSO at room temperature, so the PEU was characterized by elevated ^1^H NMR, but the signal peak resolution was still low. The proton signal peak of amino group appears at 7.02 ppm in Figure 3; the proton signal peak of terminal hydroxyl group appears at 4.34–4.32 ppm and the α–CH_2_ signal peak of –O– group on the carbamate group locates at 3.90 ppm; the methylene proton signal peak at 3.37–3.35 ppm is derived the terminal hydroxyl group attached to methylene, the 2.93 ppm is arising from α–CH_2_ proton signal peak linked to the amino group. In addition, the peaks at 5.7 ppm and 6.7 ppm were the –NH proton on the urea group. Our previous work gives were two possible reasons for the formation of urea groups [34]. First, during the transurethanization process, the reaction between terminal hydroxyl and carbamate group also possibly led to a terminal amine and a terminal carbonate group through the “tetra-hedral intermediate” transition state of the addition-elimination mechanism. It can be found that the signal peaks of the urea group in the prepolymer were obvious. The urea group mainly originates from the side reaction at high temperature. However, it was difficult to calculate the content of the urea group because the α–CH_2_ proton peak at 2.93 ppm linked to the amino group does not split. In Figure 3, the position of the –NH proton signal peaks on the carbamate and urea groups has shifted due to the use of an elevated hydrogen spectrum. The peak at 6.45 ppm is the proton signal peak of the amino group on the carbamate, and the signal located at 5.30 ppm was the –NH proton signal peak on the urea group. The positions of the other signal peaks belonging to the hard segment did not change. The peak located at 3.38 ppm arises from α–CH_2_ proton signal peak attached to the ether bond on the soft segment. The NMR hydrogen spectrum indicated that the synthesized polyether polyurethane contains carbamate hard segment and polyether soft segment.

Figure 4 showed the ATR-FTIR spectra of PEU. It can be observed from Figure 4 that the peaks located at 3326 cm^−1^, 2928–2792 cm^−1^, 1545 cm^−1^ and 1259 cm^−1^ are the stretching vibration peak of amino group, the stretching vibration peak of methylene group, the bending vibration peak of amino group and asymmetric stretching vibration peak of the C–O–C ether bond were located at, respectively. The carbonyl region at 1730–1660 cm^−1^ was magnified, and it was found that it consists of three peaks, namely, 1720 cm^−1^ of the free carbonyl peak on the carbamate group at, 1682 cm^−1^ of carbonyl peak on the carbamate group with hydrogen bonding, and 1662 cm^−1^ of the carbonyl peak on the urea group. Obviously, the peaks of the four samples in this region were different. Among the four samples, intensity of the free carbonyl peak on the carbamate group of PEU4 at 1720 cm^−1^ is highest, suggesting that relative amount of hydrogen bonding is low. On the other hand, the intensity of the urea peak at 1662 cm^−1^, PEU4 was relatively high. As far as PEU6, PEU8 and PEU10 are regarded, they had relatively high intensity of the peak at 1682 cm^−1^ derived from the carbonyl peaks on the carbamate group with hydrogen bonding, indicating that the relative amount of hydrogen bonding between their hard segments was high.

ATR–FTIR spectral data can be applied to calculate the degree of hydrogen bonding of the carbonyl group on the hard segment carbamate and assess the degree of microphase separation of the polyurethane. To quantitatively analyze the ATR-FTIR spectral data, the peaks in ATR-FTIR carbonyl region of PEU were differentiated and fitted using Peak fit software. After baseline calibration, the carbonyl region of the ATR-FTIR spectra of all four samples can be divided into five Gaussian fitted peaks. The maximum fitting error was less than 0.5% and the correlation coefficient was all greater than 0.995. The fitting results were shown in Figure 5, and the percentage of the area of each fitted peak and their attributions were listed in Table 2. The peaks at 1644 cm^−1^ and 1662 cm^−1^ belong to the carbonyl groups on the hydrogen-bonded urea groups in the ordered and disordered phases, respectively; the peaks at 1684 cm^−1^ and 1702 cm^−1^ belonged to the carbonyl groups on the hydrogen-bonded carbamates in the ordered and disordered phases, respectively; the peaks at 1720 cm^−1^ belonged to the free carbamate carbonyl groups.

According to the study of Niemczyk et al. [35], the proportion of the bonded carbamate carbonyl groups *X*_b_ can be calculated by the following equation:(1)$$X_{b}=\frac{C_{b}}{C_{b}+C_{f}}=\frac{A_{b}/\varepsilon_{b}}{A_{b}/\varepsilon_{b}+A_{f}/\varepsilon_{f}}=\frac{A_{b}}{{k^{\prime }A}_{f}+A_{b}}\left(k^{\prime }=\varepsilon_{f}/\varepsilon_{b}=1.2\right)$$
where A, C, and ε represent the absorption peak intensity, component concentration, and molar extinction coefficient, respectively, and the subscript b and f represent the bonded and free states, respectively. A_b_ refers to the sum of the areas of the peaks 1684 cm^−1^ and 1702 cm^−1^, while A_f_ is the area of the peak located at 1721 cm^−1^. When *X*_b_ is obtained, the mass fraction W’, the hard segment fraction dissolved in the soft segment region, can be calculated as follows:(2)$$W^{\prime }=\frac{\left(1-X_{b}\right)f}{\left(1-X_{b}\right)f+\left(1-f\right)}$$
where f refers to the mass fraction of the hard segment of polyurethane, while the mass fraction of the mixed phase (MP) can be calculated by the following equation:(3)$$\mathrm{MP}={\mathrm{fW}}^{\prime }$$

The results of the ATR-FTIR fitting calculations for the four samples were listed in Table 2. From the data in the table, it could be found that PEU4 differs significantly from other three samples. Its urea peak area accounts for 25% of the total carbonyl region, which was the highest among the four samples. It should be noted that the ratio of the carbonyl peak area of the ATR-FTIR urea group cannot represent the actual urea group content, because the actual urea group content has to be divided by the molar extinction coefficients of the urea and carbamate groups and the ratio of the molar extinction coefficient of the carbamate group to urea group was unknown. However, its area ratio could be used to compare the relative content of the urea groups of PEU. The proportion of bonded carbamate carbonyl groups *X*_b_ of PEU4 was 42.73%. It is lowest among the four samples, indicating the lowest relative amount of hydrogen bonding between the hard segments of PEU4. The mixed phase content MP was 5.91%. It is the highest among the four samples, suggesting its lowest degree of phase separation. The higher content of urea groups might be the main reason for the weak hydrogen bonding between PEU4 hard segments. The presence of a large number of urea groups on the molecular chains of the hard segments could disrupt the formation of ordered hydrogen bonding plane between the carbamate groups and reduce the strength of the hydrogen bonds between hard segments, which in turn results a lower degree of crystallization of the hard segments. In addition, the urea groups may form hydrogen bonds with the ether oxygen groups on the soft segments, and reduce the degree of phase separation [36]. The three other samples showed relatively similar results, whose percentage of urea group peak area is around 12% and the proportion of the bonded carbamate carbonyl *X*_b_ is 70–73%, and the mixed phase content is 3.0–3.3%. The effects of urea group content, strength of hard segment hydrogen and the mixed phase content on the thermal properties and crystallization behavior of the polymers will be elaborated in this work in detail.

### 3.2. Thermal Properties and Crystallization Behavior of PEU

The thermal properties of the PUDLs and PEUs were determined by DSC, and the heating and cooling curves for each sample were shown in Figure 6 and Figure 7. Thermal performance parameters of the soft and hard segments and four polyether polyurethanes are listed in Table 3. It can be found from Table 3 that the four synthesized hard segments are all crystalline prepolymers, and PUDL4 has the highest melting point of 164 °C and an enthalpy of melting of 66 J/g. The hard segment prepared from 1,6-hexamethylene diisocyanate (HDI) and BDO reported in the literature has a melting point of 167 °C and an enthalpy of 119 J/g during second heating [37]. The difference in melting point is not obvious but the difference in enthalpy of melting is very evident, suggesting that the enthalpy of melting and crystallization is very low when BHC is applied to synthesize hard segment. The melting points of PUDL6, PUDL8 and PUDL10 were 152 °C, 153 °C and 148 °C, respectively, which were similar with the reported value in the literature, but their melting enthalpies were significantly smaller than those of the corresponding prepolymers derived from HDI route [38,39]. Therefore, it can reach a conclusion that the degree of crystallinity of the polyurethane hard segment prepolymer prepared from ester exchange using BHC was significantly lower than that of the prepolymer prepared from conventional HDI. This might be due to the formation of urea groups as a by-product under high temperature and high vacuum conditions of the polycondensation reaction, resulting in reduction in the strength of hydrogen bonds between the hard segment molecular chains. Soft segment PTMG has a glass transition temperature at −83 °C and a melting point at 24 °C. The glass transition temperature of the soft segment was far below room temperature, which endows polyether polyurethanes to have good properties at low temperature.

The heating curves for the polyether polyurethanes obtained by condensation of the soft and hard segments were shown in Figure 7. All four samples showed one glass transition temperature (*T*_g_) of −76–−79 °C, which is close to that of PTMG soft segments (−83 °C). PEU4 had only one melt peak, which is attributed to the melting of the soft segment, while PEU6 and PEU8 had two melt peaks, which are arising from melting of the soft segment and hard segment, respectively. Besides the melt peak of the soft segment, PEU10 had two melt peaks, which probably belong to the melt peaks of the hard segment with different degrees of polymerization [34]. Based on the enthalpy of melting of hard segments, it could be speculated that PEU8 and PEU10 have higher melting enthalpy and crystallinity of hard segments. The hard segment of PEU4 is unable to crystallize, which might be due to its high urea group content. The urea group forms strong hydrogen bonding with the soft segment, resulting in a lower degree of phase separation and difficulty for the hard segment to aggregate to form an ordered structure. It can also be confirmed by the ATR-FTIR fitting results mentioned beforehand, which indicates that degree of hydrogen bonding in the hard segment of PEU4 is low and content of the mixed phase of PEU4 is high. The melting point and melting enthalpy of the soft segments of PUDL8 and PUDL10 was relatively low, which is due to the high degree of crystallization of their hard segments and subsequent a narrower space for the crystallization of soft segments. During the cooling curve of polyether polyurethanes, the crystallization temperature of the soft segment was well below room temperature, thus making the soft segment of polyether polyurethanes amorphous at room temperature and endowing the material excellent flexibility.

The WAXD spectra of the soft and hard segments are shown on the left in Figure 8, and WAXD pattern of the PEU on the right of Figure 8. It can be found that both the soft and hard segments have strong characteristic diffraction peaks and are crystallizable prepolymers, which agrees well with the DSC results. The diffraction peak shapes of PUDL4 and PUDL6 are very similar, and their diffraction peaks are at 19.6°, 22.0° and 24.1°, respectively, which were consistent with the previous reports, belonging to the trigonal crystal system. PUDL8 and PUDL10 have similar diffraction peak shapes, with diffraction peak positions of 19.6°, 21.4°, 22.6° and 24.1°, respectively, and also attribute to the trigonal crystal system [40]. In the WAXD pattern of the PEU on the right, the characteristic diffraction peaks of the soft segment disappear for all four samples, which is due to the melting and amorphous state of the soft segments at room temperature. The strong hydrogen bonding between the molecular chains of the hard segments facilitates the crystallization of the hard segments. These hydrogen bonding planes are stacked layer by layer in the direction of the molecular chain, forming a crystal lattice, whose crystal structure is similar to that of polyamide and belongs to the trigonal crystal system. The ATR-FIIR results show that PEU4 has the weakest hydrogen bonding between the hard segment molecular chains, while PEU8 has the strongest hydrogen bonding between the hard segment molecular chains. The WAXD pattern of PEU4 has only one large amorphous peak, indicating that it was an amorphous sample. For the three other samples, crystallinity degree of the hard segment of PEU6 is lower and amorphous fraction is higher, so only a weak characteristic diffraction peak of hard segment at 24.1° can be found. In contrast, the hard segments of PEU8 and PEU10 has higher degree of crystallinity, and the characteristic diffraction peaks of the hard segments were more pronounced. The WAXD results were well consistent with the DSC results, both suggesting that the degree of crystallinity in the hard segment of PEU4 is lowest and the degree of crystallinity in the hard segments of PEU8 and PEU10 is higher.

### 3.3. Morphology of PEU

SAXS is regarded as a useful method to investigate the phase separated structure of PEU. Figure 9 shows the small-angle diffraction curves of the four samples. It could be observed from Figure 9 that there is a weak scattering peak that belonged to PEU4, while the three other samples have distinct small-angle characteristic peaks formed by the periodic structure. During the cooling process of PEU, the hard segment crystallizes first and induces phase separation to form a crystalline hard segment region and an amorphous mixed-phase region containing molten soft segment and amorphous hard segment. The long periods obtained in the SAXS may originate from the lamellar structure of the hard segments or periodic structure formed by microphase separation. The value of the long period could be calculated from the q_max_ corresponding to the peak characteristic peak intensity, according to the Bragg formula: L = 2π/q_max_. Values of long periods of PEU are listed in Table 4. It can be seen from Table 4 that PEU6, PEU8 and PEU10 have long periods around 18 nm. According to Xiang et al., this periodic structure may originate from the lamellar structure of the hard segment [41]. The hard segment of PEU4 does not crystallize, and the soft and hard segments are highly compatible, so no characteristic diffraction peak appears on the small angle diffraction pattern.

The morphology of PEU was further studied by AFM, and the resulting height im-ages of the PEU films are demonstrated in Figure 10. PEU4 had a high degree of compatibility between the hard and soft segments, and no obvious phase separation pattern; the hard segments of PEU6, PEU8, and PEU10 aggregate into fibrous hard segment regions, and the hard segment “fibers” of PEU8 and PEU10 are arranged in a very regular and orderly manner. In fact, they were part of a spherical crystal formed by the hard segments. As shown in Figure 11, the spherical crystal structure formed by PEU8 and PEU10 hard segments could be found in the area with a side length of 10 μm. The size of these spherical crystals was relatively small, around 1 µm. No spherical crystal structures can be observed for PEU4 and PEU6.

### 3.4. Tensile Strength of PEU

The mechanical properties of PEU were evaluated by tensile test. All PEU samples exhibited excellent processing properties during the injection molding process. Their tensile strength, tensile modulus and elongation at break were listed in Table 5. The results showed that the tensile strength of PEU was in the range of 10.7 to 15.1 MPa, the elastic modulus ranges from 6.8 to 12.9 MPa, and the elongation at break was in the range of 511–1262%. PEU demonstrates excellent flexibility due to the good flexibility of polyether soft segment. The tensile strength and elastic modulus of polyether polyurethane are closely related to the degree of crystallinity of hard segment. The PEU4 hard segment does not crystallize, demonstrating lower tensile strength and elastic modulus. Both tensile strength and elastic modulus increase with increasing length of hard segment, and reach the highest value for PEU8. PEU8, whose hard segment has the highest degree of crystallinity, displays highest tensile strength and elastic modulus. As compared with PEU8, relative lower tensile strength and elastic modulus of PEU10 is related to its lower molecular weight and higher MP.

## 4. Conclusions

In this work, hard segments of polyurethane were prepared from hexamethylenedicarbamate and linear diols of different lengths, and a series of polyether polyurethanes were subsequently prepared from polycondensation of synthesized hard segment and PTMG2000 soft segment. Their structure was characterized by ATR-FTIR, ^1^H NMR and GPC. The intensity of hard segment hydrogen bonding and the degree of microphase separation were evaluated by peak fitting in the carbonyl region of ATR-FTIR spectra. The thermal and crystallization properties were characterized by DSC and WAXD. The morphology was characterized by SAXS and AFM. The results shows that the urea-group formed by side reaction affects the degree of crystallization of hard segments by influencing the hydrogen bonding of hard segments. The degree of crystallization of hard segments influences thermal and mechanical properties of polymer. The urea-group content was corelated to the carbon chain length of the diols used for the synthesis of hard segment. When the hard segment was synthesized from butanediol, the content of urea group was highest, and the hard segment in the polymer could not crystallize, so PEU4 has the lowest tensile strength and elastic modulus. When octanediol and decanediol were applied, urea-group content of hard segment is lower, and resulting polyether polyurethane has higher crystallization degree and tensile strength. The hard segment affects the side reaction, crystallinity, thermal properties and mechanical properties of non-isocyanate polyether-based polyurethane, and PEU8 has a low degree of side reaction, high crystallinity and good properties and mechanical properties.

## Figures and Tables

**Figure 1 polymers-14-02039-f001:**
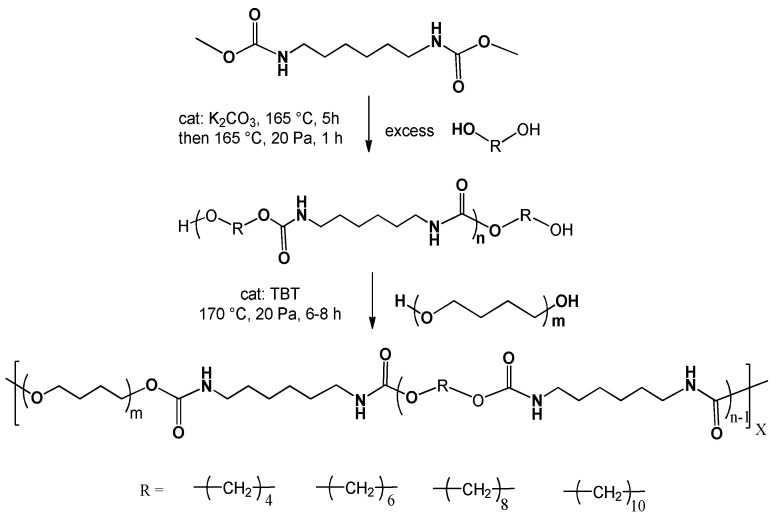
Scheme for the preparation of PEU based on different hard segments.

**Figure 2 polymers-14-02039-f002:**
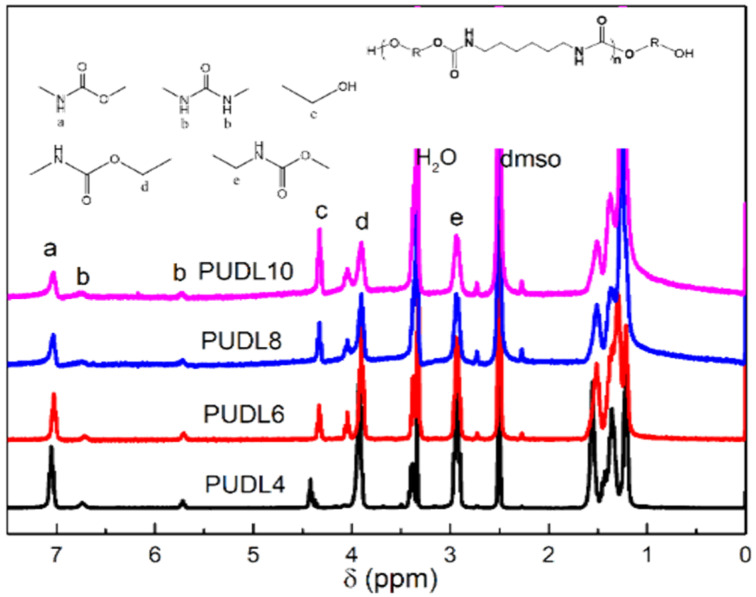
^1^H NMR spectra of PUDL.

**Figure 3 polymers-14-02039-f003:**
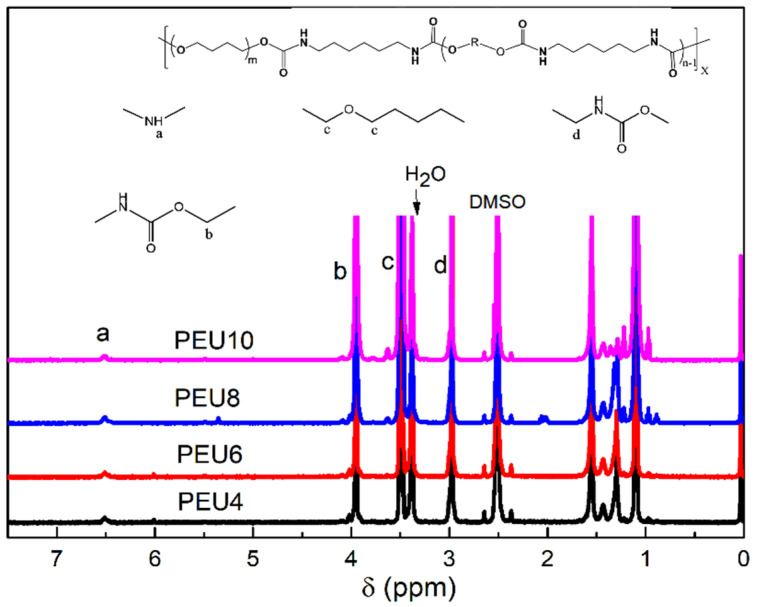
^1^H NMR spectra of PEU.

**Figure 4 polymers-14-02039-f004:**
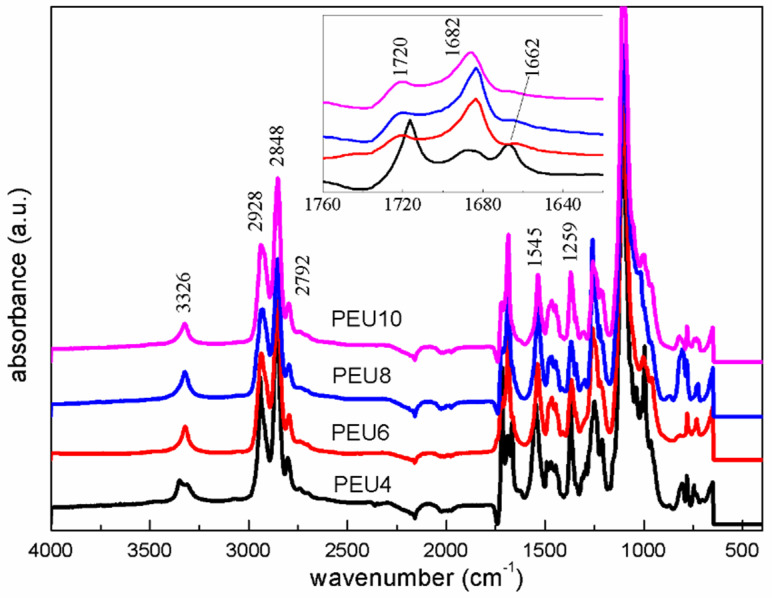
ATR–FTIR spectra of PEU.

**Figure 5 polymers-14-02039-f005:**
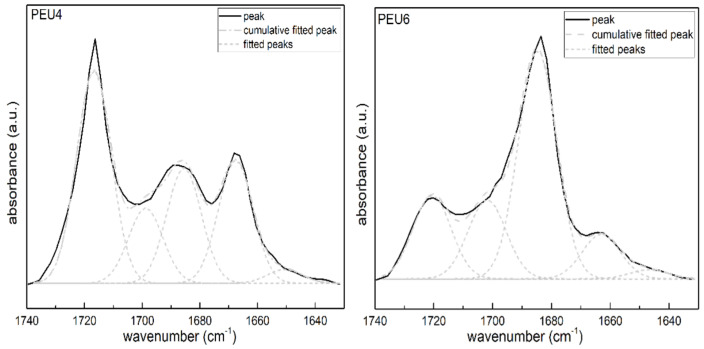
Infrared carbonyl region split peak fit of PEU.

**Figure 6 polymers-14-02039-f006:**
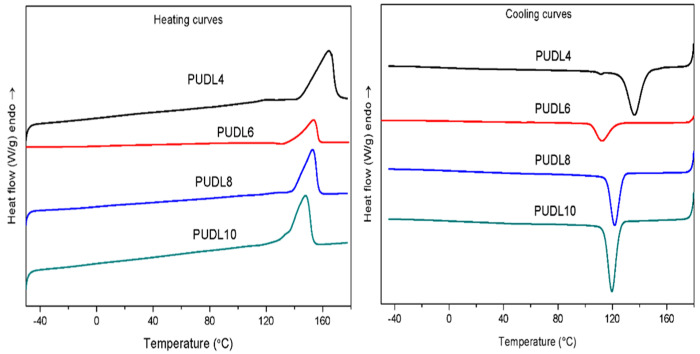
DSC heating scans (**left**) and cooling scans (**right**) of PUDL.

**Figure 7 polymers-14-02039-f007:**
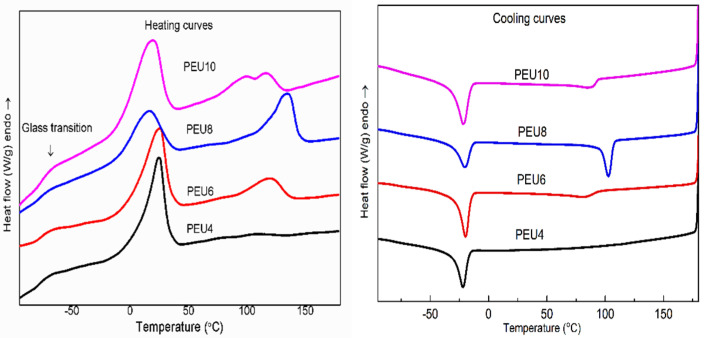
DSC heating scans (**left**) and cooling scans (**right**) of PEU.

**Figure 8 polymers-14-02039-f008:**
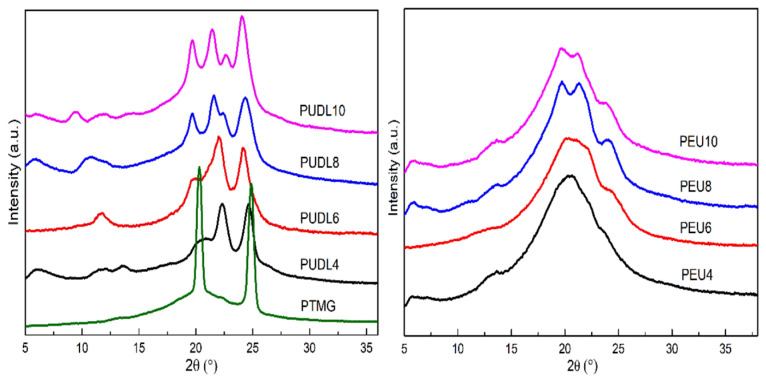
WAXD patterns of PTMG, PUDL and PEU.

**Figure 9 polymers-14-02039-f009:**
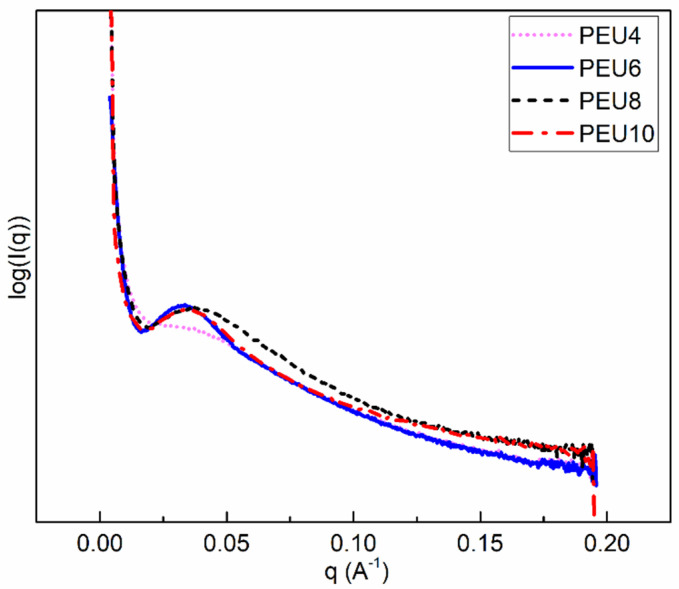
SAXS patterns of PEU.

**Figure 10 polymers-14-02039-f010:**
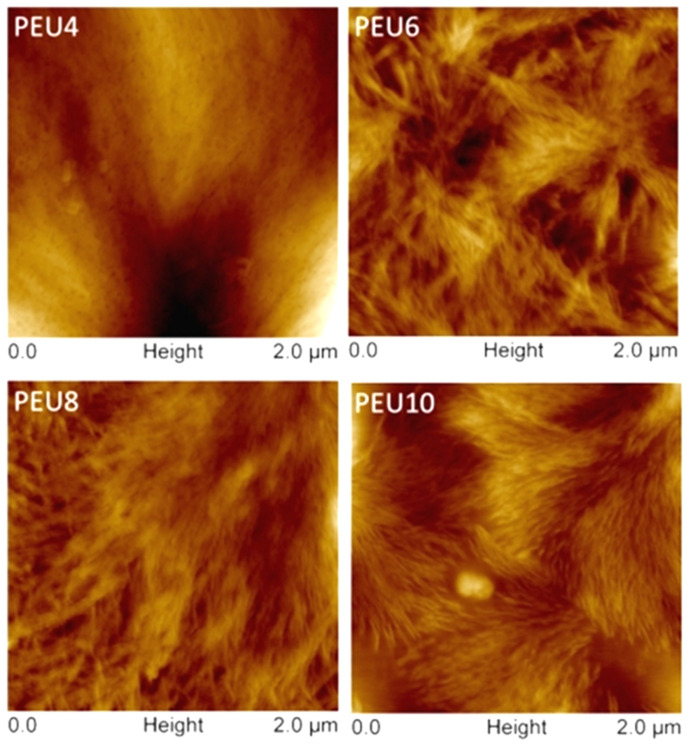
Height images of PEU (Size: 2 μm).

**Figure 11 polymers-14-02039-f011:**
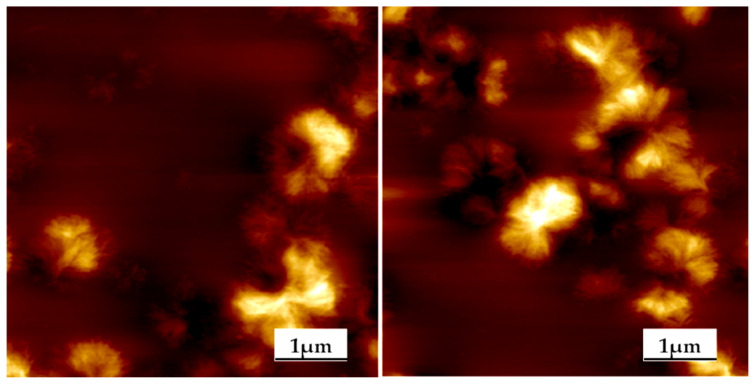
Spherical crystal structure of PEU8 and PEU10 hard sections.

**Table 1 polymers-14-02039-t001:** Compositions and molecular weights of PEU.

	Hard Segment Diols	Weight Fraction of Hard Segment	*M*_n_ (g/mol)	Đ ^a^
PEU4	Butanediol	30%	54,000	1.7
PEU6	Hexanediol	30%	61,000	1.9
PEU8	Octanediol	30%	59,000	2.3
PEU10	Decanediol	30%	53,000	2.1

^a^ Dispersity value.

**Table 2 polymers-14-02039-t002:** PEU infrared carbonyl region fitting peak attributions and proportions.

	PEU4	PEU6	PEU8	PEU10
1644 urea (bonded, ordered)	2.62%	2.88%	2.01%	2.22%
1662 urea (bonded, disordered)	22.72%	9.89%	10.05%	10.07%
1684 urethane (bonded, ordered)	21.35%	45.32%	51.50%	47.53%
1702 urethane (bonded, disordered)	13.92%	19.00%	17.72%	19.68%
1721 urethane (free)	39.39%	22.95%	18.73%	20.51%
$X_{b}$	42.73%	70.02%	75.49%	73.2%
W’	19.71%	11.38%	9.51%	10.3%
MP	5.91%	3.41%	2.85%	3.09%

**Table 3 polymers-14-02039-t003:** Thermal parameters of PUDL, PTMG and PETU.

	*T*_g_ (°C)	*T*_m1_ (°C)	Δ*H*_m_ (J/g)	*T*_m2_ (°C)	Δ*H*_m_ (J/g)	*T*_c1_ (°C)	*T*_c2_ (°C)
PUDL4	-	-	-	164	66	-	136
PUDL6	7	-	-	152	79	-	112
PUDL8	0	-	-	153	59	-	122
PUDL10	-	-	-	148	68	-	120
PTMG	−83	24	98	-	-	9	-
PEU4	−79	24	32	-	-	−22	-
PEU6	−76	24	33	120	9	−22	82
PEU8	−79	13	17	135	14	−21	103
PEU10	−76	15	28	97, 118	12	−22	87

**Table 4 polymers-14-02039-t004:** Domain parameters of PEU.

	PEU4	PEU6	PEU8	PEU10
q_max_ (Å^–1^)	0.0333	0.0334	0.0361	0.0331
d (nm)	18.9	18.8	17.4	18.9

**Table 5 polymers-14-02039-t005:** Mechanical properties of PEU.

Sample	σ_b_ (MPa)	E (MPa)	ε_b_ (%)
PEU4	10.7 ± 0.8	6.8 ± 0.8	1262 ± 59
PEU6	12.0 ± 1.0	7.3 ± 1.0	1252 ± 62
PEU8	15.1 ± 1.2	12.9 ± 0.9	1001 ± 110
PEU10	13.6 ± 1.2	9.1 ± 2.0	511 ± 55

## Data Availability

Not applicable.

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
