# Peer review of "A Non-Isocyanate Route to Poly(Ether Urethane): Synthesis and Effect of Chemical Structures of Hard Segment"

_polymers, 2022, doi:10.3390/polym14102039_

Round 1

Reviewer 1 Report

The manuscript titled as ‘A Non-isocyanate Route to Poly(ether urethane): Synthesis and Effect of Chemical Structures of Hard Segment’ focuses on the preparation of non-isocyanate polyether polyurethanes (PEU) by using the dicarbamate route. A systematic study was performed to investigate the effect of diol structure on the structure-property-relationships of the synthesized PEUs. To do so, various short diols of different lengths were used for the synthesis. The resulting polymers were analyzed by means of various analytical tools such as NMR, FTIR and DSC. Discussions were made to explain the measured differences between the polymers.

  • Related to the thermal properties of the polymers given from DSC scans:

What is the relative melting peak of a hard segment, considering the fact that the PEU 6-8-10 displayed 2nd melting signals? It would be better to compare it to previous work or literature and make the necessary discussions.

  • There is an attribute to AFM analysis for microphase characterization of the synthesized polymers. But I could not read about it in the manuscript except some attributions in abstract, experimental section and conclusions. This issue needs to be cleared out and necessary changes need to be done in the manuscript. AFM would display the differences in microphase separation to support the suggested phase mixing phenomenon of PEU-4 polymer and explain more the observed thermal and ATR-FTIR properties of the polymers.

Reviewer 2 Report

This paper presented an investigation of the chemical structures' impact on hard segments in polyether urethane. The authors first provided a brief introduction. Then, the materials, synthesis, and characterization procedures were discussed. The authors presented adequate results to support their scientific discoveries. However, some minor typos and errors were presented in the current draft. The reviewer suggests revising this paper before accepting it for publication. 

  1. The novelty of this paper can be better explained in the first section. Please clearly address what are the new discoveries of this paper in the last paragraph of the revised paper.
  2. In Figure 3, the number in the bottom left corner was missing. Please correct this figure. 
  3. In section 3.4, did the authors prepare their tensile samples and conducted the experiments following any standards? If so, please point out which standard. If not, please explain why.  

Reviewer 3 Report

Please find enclosed my comments.

Round 2

Reviewer 1 Report

Dear authors,

Thank you for the corrections and additions to the manuscript. I suggest the acceptance of the manuscript as it is.

Best regards.

Reviewer 3 Report

Dear authors,

it is a pity the TGA analysis cannot be conducted due to corona situation because it would have been interesting to see possible differences during TGA analysis of the different NIPUs. Nonetheless I recommend the manuscript for publication without further modification.